# Is studying medicine good for your health? Long-term health outcomes of a cohort of clinical medicine graduates in England and Wales in the ONS Longitudinal Study

Nicola Shelton  ,[1] Oliver Duke-Williams,[2] Laura van der Erve,[3] Jack Britton,[4] Wei Xun[1]

[1]Department of Epidemiology and Public Health, Centre for Longitudinal Study Information and User Support (CeLSIUS), UCL, London, UK
[2]Department of Information Studies, UCL, London, UK
[3]Institute for Fiscal Studies & Department of Economics, UCL, London, UK
[4]Institute for Fiscal Studies, London, UK

**Correspondence to**
Professor Nicola Shelton;
n.shelton@ucl.ac.uk

## ABSTRACT

**Objective** To quantify the potential protective effect on health associated with study of a clinical medicine degree.
**Design** Prospective population-based cohort data collected at census and linked over time: cohort born before 1976 and survived to 2011. Subgroup analysis on those who reported having a degree at 1991 census.
**Setting** England and Wales population-based, including institutions.
**Participants** 159 116 men and 174 062 women; 13 390 men with degrees and 8143 women with degrees.
**Main outcome measure** Self-reported general health in 2011 based on logistic regression analysis.
**Results** Male graduates had 92% higher odds of having good or very good health than male non-graduates after adjustment for age and socioeconomic position (CI 1.82 to 2.03). Female graduates had 85% higher odds of having good or very good health than female non-graduates after adjustment for age and socioeconomic position (CI 1.73 to 1.98). Male clinical medicine graduates had 45% higher odds of having good or very good health than male humanities graduates after adjustment for age and socioeconomic position (CI 1.09 to 1.92). Male physical sciences graduates also had higher odds of having good or very good health than male humanities graduates after adjustment for age and socioeconomic position, but life sciences and social science graduates did not. There were no significant differences by degree subject for women.
**Conclusions** Male graduates in clinical medicine have higher odds of good self-reported health. Knowledge of medicine may confer a health advantage for men above that of other degrees.

## INTRODUCTION

Several studies have shown the health benefits of education and the gradient in this. Education to a degree level confers a greater health advantage.

Freedman and Martin[1] found that education level accounted for declines in functional limitations among older Americans from 1984 to 1993; high school graduate education was

the most important in accounting for recent trends of the eight demographic and socioeconomic variables they considered.

### Strengths and limitations of this study

► The study has a large sample size (total n>1 000 000 in full panel) and is by far the largest nationally representative longitudinal data set in the UK.
► The length of follow-up available was 20 years (1991–2011 for the main census data), with life events for Office for National Statistics Longitudinal Study (ONS LS) members available until about 2 years before the current year of analysis.
► The high tracing rates contribute to the high linkage rate of LS members from census to census (88% 2001–2011).
► There were no behavioural or income data collected.
► Census is every 10 years so updates are limited.

Elo and Preston[2] found proportionate reductions in mortality for each 1-year increase in schooling in the USA at ages 35–54, comparable with those estimated for a number of European countries by Valkonen.[3] The main difference they found between USA and Europe was that in the USA reduction in mortality with years of schooling was quite similar for both men and women, while in a number of European countries male mortality reduced with educational attainment more than female mortality.

Walsemann *et al*[4] explored if later life qualifications benefited health. Among respondents with no degree, a high school diploma, or a post-high school certificate at 25 years of age, attaining at least a bachelor's degree by midlife was associated with fewer depressive symptoms and better self-rated health at midlife, compared with respondents who did not attain a higher degree by midlife.

Rogers *et al*[5] showed that educational degrees were associated with reduced risk of mortality in three cohorts of US adults aged 25 and above in 1997–2002, although they showed more marked gender differences, with associations not significant in older women and weaker in women than in men. Among men in all cohort groups, there were gradients by educational degree level in the risk of death. The overall educational degree gradient was evident in all cohorts of women, although the mortality advantages for those with postsecondary degrees were generally not as pronounced among women as among men.

More recently Buckles and Hagemann[6] found that college (university) education among white men born between 1942 and 1953 in the USA was associated with lower mortality and higher earning, and also greater reduction in smoking and higher level of physical activity. College education was also associated with higher levels of health insurance, offering a pathway to better health outcomes, but more inequality in the USA.

We highlight the role of education for several reasons. First, education is strongly associated with many health-related behaviours over the life cycle, which are frequently not measured directly in nationally representative surveys and administrative data. Further, unlike some other measures of socioeconomic status such as occupation and income, educational qualification data are straightforward to report and are generally fixed for each individual relatively early in life. In addition, higher education participation in the UK increased from 3.4% in 1950 to 8.4% in 1970, to 19.3% in 1990.[7]

There has also been considerable work looking at the earnings returns to different degree subjects,[8 9] and also work looking at the wider returns to attending higher education, including health outcomes.[10 11] However, there has been very little investigation in the health returns by different degree subjects. The UK government is increasingly focused on the returns to different degrees as government subsidies of different subject areas have increased significantly[12]; understanding these wider returns is therefore highly important.

The research looks at the health outcomes of adults by which degree they studied. Self-rated health correlates strongly with clinical assessments of morbidity and subsequent mortality in many studies. In the Office for National Statistics (ONS) Longitudinal Study (LS) specifically, a strong association has been shown between reporting of fairly good health and not good health combined, compared with good health, with mortality.[13] The hypothesis was that clinical medicine would confer health advantage. Graduates in humanities have lower salaries and lower employment rates in the UK than graduates in medicine and science,[14] yet there are no studies of how this related to health outcomes. This is the first study to consider the health benefits of studying different degree subjects.

We hypothesise that clinical medicine graduates will have higher odds of good or very good self-reported health compared with fair, bad or very bad health combined.

## METHODS

The ONS LS comprises people born in one of four selected dates of birth and so makes up about 1% of the total population in England and Wales. Data are linked for five successive censuses starting in 1971; new LS members enter the study through birth and immigration and existing members leave through death and emigration, but their data are retained.[15] The LS is representative of the whole population of England and Wales, including those in non-private households. The LS has minimal bias due to non-response or attrition, as census coverage is good and rates of linkage are high. The high tracing rates contribute to the high linkage rate of LS members from census to census (88% 2001–2011).[16] Response rates to the 2011 census were very high relative to other national censuses and sample surveys, cohort and panel studies at 94%.[17]

Adults with post-age 18 qualifications were asked the titles, subjects, awarding institutions and year in the 1991 census. These pre-1991 graduates include anyone with a degree prior to the 1991 census. We have restricted this sample to those born before 1976 (to exclude children who may have been erroneously assigned a higher educational qualification) and survived to 2011 census. The qualifications were grouped as part of census data processing in 1991 by ONS into 111 subjects. The authors grouped 110 subject areas into four 2021 Research Excellence Framework main panel subject areas: A (life sciences), B (physical sciences), C (social sciences) and D (humanities), with clinical medicine removed for the basis of this analysis from life sciences and coded as a separate category.[18] The vast majority of graduates had one degree only. However, a small proportion had multiple degrees, and of these a small number of people were recoded as having a degree in clinical medicine based on later qualifications. All other graduates were coded by their first degree awarded prior to the 1991 census. Degrees awarded after 1991 by subject were not considered as this question was not asked in subsequent censuses.

Work status variables were collected at the 2011 census and used to adjust as a proxy for income as this is not collected in the census. Respondents completed a tick box of options used to determine their participation in paid work in the labour market in the week preceding each census. Working status in 2011 with those respondents considered to be 'in work' (this included working, on temporary sick leave, maternity leave, holiday or about to take up a job) with occupational social class based on the National Statistics Socio-economic Classification with four categories was used as a risk factor for analysis. The categories were managerial, administrative and professional occupations; intermediate occupations, routine and manual occupations; and never worked and unemployed combined.

Demographic and socioeconomic indicators in 2011 were included as potential covariates. Demographic variables included age and age-squared. The results are presented separately by sex.

**Table 1** Sample characteristics

Cohort born before 1976 and survived until 2011 census completion who reported having a higher educational degree in 1991 census by NS-SEC and mean age

| | Higher occupations | Intermediate occupations | Lower occupations/none | Higher occupations % | Mean age |
|---|---|---|---|---|---|
| **Men** | | | | | |
| D (humanities) | 573 | 960 | 251 | 32 | 62 |
| A (life sciences) | 701 | 408 | 115 | 57 | 61 |
| B (physical sciences) | 2853 | 1837 | 553 | 54 | 62 |
| C (social sciences) | 2212 | 1758 | 432 | 50 | 62 |
| Clinical medicine | 526 | 26 | 12 | 93 | 66 |
| Missing degree subject | 18 | 14 | 10 | 43 | 61 |
| **Women** | | | | | |
| D (humanities) | 463 | 1877 | 347 | 17 | 60 |
| A (life sciences)* | 478 | 540 | 88 | 43 | 58 |
| B (physical sciences) | 302 | 519 | 74 | 34 | 56 |
| C (social sciences) | 844 | 1882 | 261 | 28 | 57 |
| Clinical medicine | 278 | 26 | 10 | 89 | 62 |
| Missing degree subject | Suppressed | Suppressed | Suppressed | Suppressed | 54 |

Data source: ONS LS; analysis: authors' own.
*excluding clinical medicine.
NS-SEC, National Statistics Socio-economic Classification.

Respondents were asked about self-rated health: 'How is your health in general?' The outcome measure was good health and very good health combined, compared with poor (fair, bad and very bad health combined).

### Patient and public involvement

This research was done without patient involvement. Patients were not invited to comment on the study design and were not consulted to develop patient relevant outcomes or interpret the results. Patients were not invited to contribute to the writing or editing of this document for readability or accuracy.

### RESULTS

The majority of clinical medicine graduates (93% of men and 89% of women) were employed in higher occupational classifications. This is compared with 32% of male graduates in humanities subjects and 17% of female graduates (table 1). There were small differences between the mean age of the groups of graduates analysed. The mean age of male and female clinical medicine graduates in the sample was higher than that of other male and female graduate groups, respectively. The mean age of female graduates was lower than that of male graduates by 2–7 years depending on the degree subject (table 1).

Male graduates had 92% higher odds of having good or very good health than male non-graduates after adjustment for age and socioeconomic position (CI 1.82 to 2.03). Female graduates had 85% higher odds of having good or very good health than female non-graduates after adjustment for age and socioeconomic position (CI 1.73 to 1.98) (table 2).

Male clinical medicine graduates had 45% higher odds of having good or very good health than humanities graduates after adjustment for age and socioeconomic position (CI 1.09. to 1.92). Male physical sciences graduates also had higher odds of having good or very good health than humanities graduates after adjustment for

**Table 2** Odds of having good or very good health in 2011 by degree status in 1991: cohort born before 1976

| | Men, n=159116 | | | | Women, n=174062 | | |
|---|---|---|---|---|---|---|---|
| | OR | p value | CI | | OR | p value | CI |
| Does not have a degree in 1991 | 1.00 | | | | 1.00 | | |
| Has a degree in 1991 | 1.92 | <0.001 | 1.82 to 2.03 | | 1.85 | <0.001 | 1.73 to 1.98 |

Adjusted for age, age-squared and socioeconomic status (NS-SEC); constant not shown.
Data source: ONS LS; analysis: authors' own.
NS-SEC, National Statistics Socio-economic Classification.

**Table 3** Odds of having good or very good health in 2011 by degree attained by 1991: cohort born before 1976

| Subject | Men, n=13390 | | | Women, n=8143 | | |
|---|---|---|---|---|---|---|
| | OR | p value | CI | OR | p value | CI |
| D (humanities) | 1.00 | | | 1.00 | | |
| A (life sciences)* | 1.16 | 0.163 | 0.94 to 1.44 | 0.95 | 0.663 | 0.77 to 1.18 |
| B (physical sciences) | 1.24 | 0.006 | 1.06 to 1.44 | 0.82 | 0.086 | 0.66 to 1.03 |
| C (social sciences) | 1.07 | 0.371 | 0.92 to 1.25 | 0.89 | 0.140 | 0.76 to 1.04 |
| Clinical medicine | 1.45 | 0.011 | 1.09 to 1.92 | 1.10 | 0.605 | 0.76 to 1.60 |
| Missing degree subject | 1.06 | 0.889 | 0.46 to 2.48 | 0.86 | 0.773 | 0.32 to 2.32 |

Adjusted for age, age-squared and socioeconomic status (NS-SEC); constant not shown.
Data source: ONS LS; analysis: authors' own.
Subjects grouped into 2021 REF panel classes (A–C) compared with humanities (D).
*excluding clinical medicine.
NS-SEC, National Statistics Socio-economic Classification; REF, Research Excellence Framework.

age and socioeconomic position, but life sciences and social science graduates did not. There were no significant differences by degree subject for women (table 3).

## CONCLUSIONS

Male graduates in clinical medicine have higher odds of good or very good self-reported health. Knowledge of medicine may confer a health advantage for men above that of other degrees. The study of medicine may both inform personal health behaviour decisions and also lead to earlier self-diagnosis through skills gained in research of clinical information and from knowing other experts in the medical field to consult. Additionally there are financial benefits of studying medicine that may explain the health advantage. Ross and Wu[18 19] found that fulfilling work and high income were very important in explaining the education–health link. As we found in previous work[8] medicine is one of the degree subjects which increase earnings the most, not only much more than the humanities and social science degrees, but also more than other sciences, and hence this could explain some of its strong positive impact on health. (This may also partly explain the association seen in physical sciences graduates.) The census however does not collect details of income. The vast majority of clinical medicine graduates were employed in higher level occupations. Given the higher mean age of clinical medicine graduates if age selection were explaining the results, we would expect this to reduce rather than increase the size of the association found, suggesting this is not the explanation.

Why these benefits were only experienced by men might be explained by the higher salaries of male clinicians or by the benefits of health-related knowledge mediating the gender differentials in health behaviours. There may also be selection bias, with men being more likely to be admitted to medical school and more likely to pursue a career in science generally than women, and with men more likely to get employment and stay in a medical profession than women. Men's careers are less likely to be affected by family and childbearing responsibilities (although as this study looks at education rather than occupation, the latter may be less of an issue). The mean age of the women in the sample was slightly younger than men. There could also be effects on health, where the educational qualification of the head of household may be more important, especially in households which are not headed by women.

Cutler and Lleras-Muney[20] found that specific factual knowledge, for example, on the harms of smoking and drinking, accounts for around 10% of the education gradient in health behaviours. We would obviously expect this specific factual knowledge to be highest among clinical medicine graduates. This could be further investigated by studying other graduates with health-related qualifications. The Medical Schools Selection Alliance details a minimum of three A levels (post-16) with qualifications usually in laboratory-based sciences and often a third science subject for application to study medicine in the UK.[21] There are no post-16 academic subjects explicitly covering human health other than vocational and technical qualifications in health and social care,[22] with human biology A level phased out in 2017.[23] Personal, social, health and economic education is a non-statutory subject in the English school curriculum in maintained schools and academies to age 16 only, although all state schools should make provision for its teaching.[24] Whether a compulsory General Certificate of Secondary Education (GCSE) and optional A level in a health-related discipline would improve the population's health remains open for debate, and to persuade medical schools whether this would form part of a suitable suite of qualifications with which to apply to medical school could also be challenging. This study has looked at graduates of medicine rather than those practising medicine. It is beyond the scope of the paper to look at how these outcomes may differ for those who study medicine but are employed in other fields, although this is a potential area for future research and gender may play an interesting role here.

Also disentangling the effects of income might be considered if data on income were in the future able to be linked to the ONS LS, perhaps as part of administrative-based censuses.

**Acknowledgements** The permission granted by the ONS to use the Longitudinal Study is gratefully acknowledged, as is the help provided by the staff of the CeLSIUS. The authors alone are responsible for the interpretation of the data.

**Contributors** NS and OD-W devised the research idea. NS, OD-W and WX collated the data and completed the analysis. LVDE and JB cowrote the literature review, in conjunction with NS and OD-W, and provided comment on the full draft text.

**Funding** CeLSIUS is funded by the Economic and Social Research Council (award refs: ES/R00823X/1, ES/V003488/1).

**Disclaimer** The authors alone are responsible for the interpretation of the data.

**Competing interests** None declared.

**Patient and public involvement** Patients and/or the public were not involved in the design, or conduct, or reporting, or dissemination plans of this research.

**Patient consent for publication** Not required.

**Ethics approval** The research project was approved by the ONS Longitudinal Study Research and Development Board (project ID: 1007013).

**Provenance and peer review** Not commissioned; externally peer reviewed.

**Data availability statement** Data may be obtained from a third party and are not publicly available. The Office for National Statistics Longitudinal Study (ONS LS) data from which this panel is drawn are available from ONS via the Secure Research Service to approved researchers with approved projects. Information and support for the LS for UK-based prospective and current users from the academic, statutory and voluntary sectors can be obtained from the Centre for Longitudinal Study Information and User Support (CeLSIUS) by emailing Celsius@ucl.ac.uk. All other users should contact the ONS Longitudinal Study Development Team (LSDT): LongitudinalStudy@ons.gov.uk. A step-by-step guide to using the LS is available from the CeLSIUS website (www.ucl.ac.uk/celsius). This work contains statistical data from ONS which are Crown Copyright. The use of the ONS statistical data in this work does not imply the endorsement of the ONS in relation to the interpretation or analysis of the statistical data. This work uses research data sets which may not exactly reproduce National Statistics aggregates.

**ORCID iD**
Nicola Shelton http://orcid.org/0000-0002-4939-1036

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
