## [Reviewer comments · BMJ Open]

This paper was submitted to a another journal from BMJ but declined for publication following peer review. The authors addressed the reviewers' comments and submitted the revised paper to BMJ Open. The paper was subsequently accepted for publication at BMJ Open.

(This paper received 2 reviews from its previous journal but only 1 reviewer agreed to published their review.)

ARTICLE DETAILS

TITLE (PROVISIONAL)	Is studying medicine good for your health? Long term health outcomes of a cohort of clinical medicine graduates in England and Wales in the ONS Longitudinal Study.
AUTHORS	Shelton, Nicola; Duke-Williams, Oliver; van de Erve, Laura; Britton, Jack; Xun, Wei

VERSION 1 – REVIEW

REVIEWER	Ian Shuttleworth QUB
REVIEW RETURNED	December,08,2019

GENERAL COMMENTS	This paper analyses the health outcomes of graduates with different disciplinary backgrounds. It uses linked census/administrative data from the ONS LS and the 2011 health question. Data of these type are good at measuring patterns and associations, less strong when dealing in some circumstances with causality. The analyse finds that medical graduates have better outcomes than those from the arts/social science disciplines. The question is novel and its answer adds to the literature. There are, however, some questions about the analysis. One concerns the outcome variable which is defined as those born before 1975 with a 2011 census record. Is there are any way to analyse those who dies before 2011? This would be a more direct measure of health (although self-reported health is a good robust measure). Another is about the framing of the paper. There is no clear hypothesis – are we dealing with an income effect, a knowledge effect or both? Or is the analysis purely descriptive? Either way, the conclusion is highly speculative and might be much more concise. As noted, these data are great for measuring trends and patterns at the population level, and weaker for detailed mechanisms/causes but if the analysis does try to test hypotheses, I wonder if the possible effects of income could be ruled in/out with analysis that either (a) interacts degree subject with the income proxy or (b) considers the income proxy alone. If the health effects for high-income medical graduates are strong and positive and those for low-income graduates are less or absent, that answer would be suggestive. Some minor questions – how many medical graduates and indeed those in other groups? What is happening with physical science graduates, an income or knowledge effect? Also, watch
--

	writing, errors like “The LS is minimal bias....” happen too often.
--	---

VERSION 1 – AUTHOR RESPONSE

Reviewer: 2

This paper analyses the health outcomes of graduates with different disciplinary backgrounds. It uses linked census/administrative data from the ONS LS and the 2011 health question. Data of these type are good at measuring patterns and associations, less strong when dealing in some circumstances with causality. The analyse finds that medical graduates have better outcomes than those from the arts/social science disciplines. The question is novel and its answer adds to the literature. There are, however, some questions about the analysis. One concerns the outcome variable which is defined as those born before 1975 with a 2011 census record. Is there are any way to analyse those who dies before 2011? This would be a more direct measure of health (although self-reported health is a good robust measure).

This is addressed in the follow up rate

Another is about the framing of the paper. There is no clear hypothesis – are we dealing with an income effect, a knowledge effect or both? Or is the analysis purely descriptive? Either way, the conclusion is highly speculative and might be much more concise. As noted, these data are great for measuring trends and patterns at the population level, and weaker for detailed mechanisms/causes but if the analysis does try to test hypotheses, I wonder if the possible effects of income could be ruled in/out with analysis that either (a) interacts degree subject with the income proxy or (b) considers the income proxy alone. If the health effects for high-income medical graduates are strong and positive and those for low-income graduates are less or absent, that answer would be suggestive. Some minor questions – how many medical graduates and indeed those in other groups? What is happening with physical science graduates, an income or knowledge effect? Also, watch writing, errors like “The LS is minimal bias....” happen too often.

We have extended the discussion to include income which is not available in the data set.

VERSION 2 – REVIEW

REVIEWER	Ian Shuttleworth Queen's University Belfast
REVIEW RETURNED	04-Jul-2020

GENERAL COMMENTS	The study considers an important issue given higher education policy especially with regard to the broader returns to education. The dataset used - the ONS Longitudinal Studies - has considerable strengths for this purpose, mainly its temporal depth, high linkage rate, and the wide suite of census information collected (although crucially not income). These strengths and weaknesses are assessed properly and the analysis is competently conducted using established methods. There are some minor presentational issues - the cohort date varies between 1975 and 1976 between Tables 1 and 2, and the way that the results analysis are explained in terms of the odds of good and then poor health make the story harder to untangle. These should be resolved as should the sentence starting "Both in that the study of medicine....." should be resolved. My most substantive comment concerns the sample design which is restricted
--

	to those born before 1975 (wouldn't someone born in 1975 only be 16 in 1991?) - with no start year to define the cohort eg as in 1940-1975 - and present in 1991 when degree subject was collected and surviving to 2011. This does not account for deaths between 1991 and 2011 and so by definition the analysis excludes people who were unhealthy (although of course there might have been some accidental deaths too). So, I think there is a selection issue here. In this light, the final column of Table 1 is crucial where the average age of medical survivors is 64 as compared to 60/61 for the other degree subject groups. This very nearly tells the whole story and more could be made of this especially if the differences in means are statistically significant (which I imagine they would be given the sample size). This would add to and strengthen the conclusions drawn from the results presented in Tables 2 and 3. Additionally, it would be very useful to have the average age at death of all graduates (by subject) who were alive in 1991. The analysis is necessarily a little muddy on whether the self-reported health differential by degree subject is the result of (a) a knowledge effect or (b) an income effect or (c) a mixture of (a) and (b). This weakness is acknowledged and whether (a) or (b) is in play is discussed speculatively but might be considered more briefly with some suggestions of a strategy to address the problem using this or another dataset.
--	--

REVIEWER	Kaustuv Bhattacharya University of Mississippi, USA
REVIEW RETURNED	23-Sep-2020

GENERAL COMMENTS	This study attempted to assess the impact of higher education, in general, on health status among a nationally representative sample of individuals in the UK. Furthermore, it also examines whether health status varies by degree subject among graduates stratified by gender. The results indicate that both male and female graduates had higher odds of having good health as compared to their non-graduate counterparts. Additionally, male graduates with clinical medicine degrees had lower odds of having poor health as compared to male humanities graduates. While the paper adds to the literature, and the authors do a good job of presenting the study findings, I would like to see an expanded discussion on the implications of the study findings for future research. My specific comments are as follows:  1. Page 3, Line 19 – should be “female non-graduates”, instead of “non-graduates”. 2. Page 5, Line 35 – what education group were graduates with multiple degrees but no clinical medicine degree allocated? 3. Page 6, Line 3 – Can the authors specify if the question about self-rated health has been validated as a measure of health status or if it has been previously used to assess self-rated health? 4. Page 6, Line 6 – Can the authors provide evidence for their decision to include those who responded “fairly good” to the self-rated health question in the poor health group? If dichotomizing is a problem, ordinal logistic regression models can also be used to assess the relationships under investigation 5. Page 7, Lines 39-42 – 95% CIs for ORs for both male social science graduates and male physical science graduates as compared to the reference group (male humanities graduates) are exactly the same (0.71 – 1.00), and include 1. Is there a reason why the authors state that male social science graduates had “marginally significant” lower odds of having poor health as compared to male humanities graduates, but physical science graduates did not.
--

	Looking at the 95% CI it seems that neither male social science graduates nor male physical science graduates had significantly lower odds of having poor health as compared to the reference group (male humanities graduates). Please explain. 6. Also, education was assessed as of degrees obtained at 1991 census, whereas health status was assessed as of the 2011 census. What if some of the respondents had obtained one or more educational degrees after 1991? Is that information available in the linked data? If yes, has it been accounted for? It may help the readers if the authors clarified it.
--	--

VERSION 2 – AUTHOR RESPONSE

Reviewer 1

The cohort date varies between 1975 and 1976 between Tables 1 and 2, This date has been checked and corrected in the tables and the text.

The results analysis are explained in terms of the odds of good and then poor health make the story harder to untangle.

>Table 3 has been amended to odds of good or very good health to match Table 2 so that the odds ratios are greater than 1 and are easier to interpret for the reader and all analysis rerun for both tables.

The sentence starting "Both in that the study of medicine....." should be resolved.

> This sentence has been amended to read: The study of medicine may both

My most substantive comment concerns the sample design which is restricted to those born before 1975 (wouldn't someone born in 1975 only be 16 in 1991?) - with no start year to define the cohort eg as in 1940-1975 - and present in 1991 when degree subject was collected and surviving to 2011. This does not account for deaths between 1991 and 2011 and so by definition the analysis excludes people who were unhealthy (although of course there might have been some incidental deaths too). So, I think there is a selection issue here. In this light, the final column of Table 1 is crucial where the average age of medical survivors is 64 as compared to 60/61 for the other degree subject groups. This very nearly tells the whole story and more could be made of this especially if the differences in means are statistically significant (which I imagine they would be given the sample size). This would add to and strengthen the conclusions drawn from the results presented in Tables 2 and 3. Additionally, it would be very useful to have the average age at death of all graduates (by subject) who were alive in 1991.

>We have excluded anyone under 16 so children who have erroneously been assigned a higher education qualification are excluded and a phrase has been added to this effect. Sample sizes have been added to Tables 2 and 3.

(to exclude children who may have been erroneously assigned a higher education qualification)

We have added a sentence in the results and the conclusion.

There were small differences between the mean age group of the groups of graduates analysed. The mean age of male and female clinical medicine graduates in the sample was higher than that of other male and female graduate groups respectively.

Given the higher mean age of clinical medicine graduates if age selection were explaining the results we would explain this to reduce rather than increase the size of the association found suggesting this is not the explanation.

Average age at death prior to 2011 is not available to the authors as the subsample for which the study has approval for access is restricted to the selected graduates alive in 2011.

The analysis is necessarily a little muddy on whether the self-reported health differential by degree subject is the result of (a) a knowledge effect or (b) an income effect or (c) a mixture of (a) and (b). This weakness is acknowledged and whether (a) or (b) is in play is discussed speculatively but might be considered more briefly with some suggestions of a strategy to address the problem using this or another dataset.

> A sentence has been added to the conclusion suggesting an alternative analysis.

Also disentangling the effects of income might be considered if data on income were in future able to be linked to the ONS LS perhaps as part of administrative based censuses.

Reviewer 2

1. Page 3, Line 19 – should be “female non-graduates”, instead of “non-graduates”.

> sentence revised

2. Page 5, Line 35 – what education group were graduates with multiple degrees but no clinical medicine degree allocated?

>: A sentence has been added to clarify this

All other graduates were coded by their first degree awarded prior to the 1991 Census

3. Page 6, Line 3 – Can the authors specify if the question about self-rated health has been validated as a measure of health status or if it has been previously used to assess self-rated health?

> addressed below in response to point 4

4. Page 6, Line 6 – Can the authors provide evidence for their decision to include those who responded “fairly good” to the self-rated health question in the poor health group? If dichotomizing is a problem, ordinal logistic regression models can also be used to assess the relationships under investigation

> A reference has been added to justify the combination of fairly good with not good health

In many studies and in the ONS Longitudinal Study specifically a strong association has been shown between reporting of fairly good or not good health with mortality (Young et al, 2010).

5. Page 7, Lines 39-42 – 95% CIs for ORs for both male social science graduates and male physical science graduates as compared to the reference group (male humanities graduates) are exactly the same (0.71 – 1.00), and include 1. Is there a reason why the authors state that male social science graduates had “marginally significant” lower odds of having poor health as compared to male humanities graduates, but physical science graduates did not. Looking at the 95% CI it seems that neither male social science graduates nor male physical science graduates had significantly lower odds of having poor health as compared to the reference group (male humanities graduates). Please explain.

> Though the 95% intervals appeared to include 1 this was due to rounding. The analysis has been revised and this point no longer applies.

6. Also, education was assessed as of degrees obtained at 1991 census, whereas health status was assessed as of the 2011 census. What if some of the respondents had obtained one or more educational degrees after 1991? Is that information available in the linked data? If yes, has it been accounted for? It may help the readers if the authors clarified it.

> We have added a sentence to clarify.

Degrees awarded after 1991 by subject were not considered as this question was not asked in subsequent censuses

VERSION 3 – REVIEW

REVIEWER	Ian Shuttleworth Queen's University Belfast
REVIEW RETURNED	15-Dec-2020

GENERAL COMMENTS	The paper deals with differences in self-reported health in 2011 by (a) educational attainment and (b) degree subject as reported in the 1991 Census. Its main value is in identifying for men the benefits of studying the clinical medical and physical sciences after making controls for demographic and socio-economic indicators. This usefully provides evidence to raise questions for further investigation. The data used are from the ONS LS census longitudinal study for England and Wales. The benefit of these data are that they are from a population-level dataset with low attrition rates with coverage from 1971 to 2011. I have reviewed an earlier version of this paper. This revision deals with many of the points that were raised at that stage. My comments are thus more about presentation than substantive issues. First, the paper needs a thorough proof read - there are too many cases like 'compared with fair, bed or very bad health combined' and 'The LS is minimal bias', and 'supressed'. Second, I assume that the tables will be more clearly formatted? Also the following text 'Adults with post age 18 qualifications were asked the titles, subjects, awarding institutions and year' needs to add 'in the 1991 Census'. Nevertheless, the paper makes a useful contribution.
---

VERSION 3 – AUTHOR RESPONSE

Regarding the helpful final amendments requested by of Reviewer 1

First, the paper needs a thorough proof read - there are too many cases like 'compared with fair, bed or very bad health combined' and 'The LS is minimal bias', and 'supressed'.

The paper has been proof read and such minor amendments made.

Second, I assume that the tables will be more clearly formatted?

I assume these will be reformatted by the journal but the acceptance of track changes has made them more readable.

Also the following text 'Adults with post age 18 qualifications were asked the titles, subjects, awarding institutions and year' needs to add 'in the 1991 Census'.

The required text has been added.